# Immunotherapy in Acute Myeloid Leukemia: A Literature Review of Emerging Strategies

**DOI:** 10.3390/bioengineering10101228

**Published:** 2023-10-20

**Authors:** Luca Guarnera, Carlos Bravo-Perez, Valeria Visconte

**Affiliations:** 1Department of Translational Hematology & Oncology Research, Taussig Cancer Institute, Cleveland Clinic, Cleveland, OH 44195, USA; bravoc2@ccf.org (C.B.-P.); visconv@ccf.org (V.V.); 2Department of Biomedicine and Prevention, University of Rome Tor Vergata, 00133 Rome, Italy; 3Department of Hematology and Medical Oncology, Hospital Universitario Morales Meseguer, University of Murcia, IMIB-Pascual Parrilla, CIBERER—Instituto de Salud Carlos III, 30005 Murcia, Spain

**Keywords:** immunotherapy, acute myeloid leukemia, CAR-T, checkpoint inhibitors

## Abstract

In the last twenty years, we have witnessed a paradigm shift in the treatment and prognosis of acute myeloid leukemia (AML), thanks to the introduction of new efficient drugs or approaches to refine old therapies, such as Gemtuzumab Ozogamicin, CPX 3-5-1, hypomethylating agents, and Venetoclax, the optimization of conditioning regimens in allogeneic hematopoietic stem cell transplantation and the improvement of supportive care. However, the long-term survival of non-M3 and non-core binding factor-AML is still dismal. For this reason, the expectations for the recently developed immunotherapies, such as antibody-based therapy, checkpoint inhibitors, and chimeric antigen receptor strategies, successfully tested in other hematologic malignancies, were very high. The inherent characteristics of AML blasts hampered the development of these treatments, and the path of immunotherapy in AML has been bumpy. Herein, we provide a detailed review of potential antigenic targets, available data from pre-clinical and clinical trials, and future directions of immunotherapies in AML.

## 1. Introduction

In the last twenty years, the knowledge of mechanisms of leukemogenesis and molecular/genetic features of acute myeloid leukemia (AML) has rapidly progressed, allowing a precise characterization of the disease and several updates on AML classification [1,2]. In the same fashion, we have witnessed breakthroughs in drug development for myeloid neoplasms. For instance, the most illustrative case is the drug-free combination of all-trans retinoic acid/arsenic trioxide for acute promyelocytic leukemia (APL), capable of curing the disease in the vast majority of patients [3,4].

The introduction in clinical practice of new drugs, such as Gemtuzumab Ozogamicin, CPX 3-5-1, hypomethylating agents (HMA) and Venetoclax (VEN) [5,6,7], the efficient use of measurable residual disease [8] (MRD), the optimization of hematopoietic stem cell transplantation (HSCT) conditioning regimens [9], and the improvements of supportive therapy led to a paradigm shift in the management of AML, allowing a therapeutic path tailored to the inherent features of both disease and patients. However, except for APL and core-binding factor AML (CBF-AML), the outcome of AML patients remains dismal. According to the MD Anderson case series, the 5-year overall survival rate still does not reach 20%, despite having doubled since early 2000 [10].

Thus, the recent and successful introduction of immunotherapies, such as antibody-based therapy, checkpoint inhibitors, and chimeric antigen receptor (CAR) strategies, in lymphoid malignancies has raised hopes for their possible use in AML as well.

Furthermore, other immunologic therapies showed efficacy in AML, and some of them are established as gold standards in different settings. The most illustrative example is HSCT: after HSC engraftment, the donor immune system is capable of targeting the residual leukemic cells, giving life to the so-called “Graft-versus-Leukemia” (GvL) effect [11]. Key characters in this setting are lymphocytes, in particular CD4-CD25-FoxP3-positive regulatory T cells (Tregs), which have been proven to avoid alloreactive reactions, prevent reactions of the graft against the host (“Graft-versus-host-disease” effect, GvHD), and foster engraftment [12]. Other proofs of the immune effects of these cells come from high-resolution in vivo imaging studies, showing co-localization of HSCs with Tregs, proving thus a role in the creation of a permissing microenvironment [13], and the efficacy of donor lymphocyte infusion as adoptive immunotherapy for relapse after HSCT [14]. 

Also, other AML treatments present immunomodulatory effects: in preclinical models, Azacytidine (AZA) showed efficacy in preventing GvHD in combination with Tregs, which resulted in resistance to the antiproliferative activity of the drug [15,16]. Furthermore, analysis of bone marrows from patients enrolled in the RAS-AZIC trial suggested an AZA-dependent reconstitution of T-cell repertoires [17]. In a similar fashion, histone deacetylase inhibitors (HDACIs) enhance the production and suppressive functions of FOXP3(+) Tregs and stimulate tumor antigenicity and targeted immune system-mediated cytotoxicity in CD8+ T cells and NK cells [18,19]. These results were observed in a suppression of GVHD context while retaining beneficial GVL effects [20]. Sorafenib, a FLT3 inhibitor, through the downregulation of the transcription factor ATF4, thereby blocking negative regulation of interferon regulatory factor 7, enhances IL-15 transcription in *FLT3*+ AML cells, which causes an increase in CD8+CD107a+IFN-γ+ T cells and is capable of eradicating leukemia cells [21].

On the other hand, recent immunologic strategies, such as peptide and dendritic cell-based vaccines, didn’t show satisfactory results, especially in active disease settings. Although immunological activity was documented, most of the experiences reported a low rate of clinical responses [22,23,24], with the notable exception of the trial by Dong et al., who showed the efficacy of a combination therapy with low-dose chemotherapy and dendritic cell-based vaccines vs. chemotherapy alone in elderly patients [25]. The vaccine’s efficacy as maintenance therapy in post-chemotherapy and post-HSCT settings provided better results, prolonging survival and eradicating MRD [26,27,28,29]. Thus, several ongoing phase I and II clinical trials are investigating their potential role in this setting [30].

For a more comprehensive discussion of these topics, we refer to specialized reviews [30,31]. In this paper, we review the mechanisms underlying the main emerging immunotherapy strategies in AML, the results of the early clinical trials, and the new directions.

## 2. Antibody-Based Therapy

Antibody-based therapies enclose differently-structured molecules able to bind an effector cell to AML blasts in order to activate the immune response [32]. One successful example of this class of drugs is Blinatumomab, a paradigm-shifting bispecific T-cell engager (BiTE), which has first been shown to prolong survival in acute lymphoblastic leukemia (ALL) [33]. Bispecific antibodies used in cancer immunotherapy have dual affinities targeting both tumor and immune cells, engaging them and promoting effective antitumoral responses. BiTE antibodies possess two single-chain variable fragment (scFv) regions, each of them with light (VL) and heavy (VH) chains, from two different monoclonal antibodies able to bind a tumor-associated antigen and CD3 receptor in T cells [34] (Figure 1A). Subsequent variations of this construct are dual affinity retargeting antibodies (DARTs), characterized by an alternative link between VL and VH chains from the two antibodies and an additional interchain disulfide bond to improve stabilization [35] (Figure 1B); bispecific killer engager (BiKE), binding NK cells instead of T cells via CD16 (Figure 1C); and trispecific killer engager (TriKE), BiKE characterized by the incorporation of an IL-15 linker [36] (Figure 1D). 

A common issue for both antibody-based therapy and CAR-T strategies is selecting the proper target in AML blasts since there are no antigens exclusively expressed by these cells. Illustratively, CD33 and CD123, targeted by the conjugate antibody Gemtuzumab Ozogamicin and the fusion protein Tagraxofusp, are also present on normal HSCs [37], while they are expressed in a high but submaximal rate (70–90%) of AML blasts. This may result, on the one hand, in an incomplete clearance of the blasts and, on the other hand, in harm to the healthy HSC compartment. In addition, there are side effects, both common to conventional AML management strategies, such as high rates of opportunistic and nosocomial infections due to cytopenia [38,39,40], and those related to specific targets, such as veno-occlusive disease, due to the expression of CD33 on hepatic Kupffer cells, observed also in patients treated with Gemtuzumab Ozogamicin [41], and cytokine release syndrome (CRS), due to activation of bystander immune cells and non-immune cells, such as endothelial cells, after binding the target antigen [42]. 

Both Gemtuzumab Ozogamicin and Tagraxofusp represented important breakthroughs in AML treatment: the first one was approved as first-line therapy in combination with daunorubicin and cytarabine in low-intermediate-risk de novo AML after the phase III ALFA-0701 trial and the metanalysis by Hills et al. [7,43], which showed a survival advantage in these subsets of patients vs. chemotherapy alone. Tagraxofusp was approved for the treatment of blastic plasmacytoid dendritic cell neoplasms (BPDCN) following the trial by Pemmaraju et al., which documented clinical responses in both de-novo and pre-treated patients [44]. Ongoing trials are investigating the efficacy of Tagraxofusp in other myeloid malignancies, including de novo and secondary AML [45,46]. 

The same targets (CD33 and CD123, respectively) were also used for most antibody-based strategies, as described in the next paragraphs. 

### 2.1. T-Cell-Binding Antibodies

T-cell-binding strategies (e.g., BiTE and DARTs) have been tested in refractory/relapsed (R/R) AML patients in several phase I trials (Table 1). The first BiTE data, of Flotetuzumab, a CD123 × CD3 DART, and AMG330, a CD33 × CD3 BiTE, were presented at the 2018 American Society of Hematology (ASH) meeting.

The first dose-escalation analysis of Flotetuzumab showed, in patients treated with the recommended Phase 2 dose (500 ng/kg per day administered as a continuous intravenous [IV] infusion), an overall response rate (ORR, including complete response [CR], complete response with incomplete count recovery [CRi, defined as platelet count < 100 × 10^9^/L and/or absolute neutrophil count < 1 × 10^9^/L], morphologic leukemia-free state [MLFS] and partial response [PR]) of 22% and a composite complete response (cCR, including CR, CRi, and MLFS) rate of 19%. Infusion-related reactions (IRR)/CRS occurred in all thirty patients treated (grade ≥ 3 in 13.3%) [47].

In 2021, the report of the dose expansion cohort was published. cCR rate was 20%, with an ORR of 24%. IRR/CRS were observed in 96% of patients (grade ≥ 3 in 4%). Grade ≥ 3 neutropenia and thrombocytopenia occurred in 2% and 6%, respectively [48]. A trial enrolling pediatric patients is currently ongoing (NCT04158739). A clinical trial evaluating a second-generation CD123 × CD3 DART, MGD024, engineered to minimize CRS while maintaining antitumor activity, is ongoing as well (NCT05362773).

The phase 1 dose escalation study evaluating AMG330 as a continuous IV infusion in patients with R/R AML showed a cCR rate of 11.4%, with an 89% treatment discontinuation rate for disease progression, adverse events, and/or patient requests. Grade ≥ 3 CRS was observed in 31.4% [49]. The study update in 2020 confirmed the low cCR rate (19%). CRS was observed in 67% of cases; ≥grade 3 in 13% [50].

In 2019, the results of AMG673, a CD33 × CD3 BiTE genetically fused to the N-terminus of a single-chain IgG Fc region (to increase the half-life of the molecule), phase I study were presented: an ORR of 22.2% with a cCR rate of 3.7% were observed. CRS occurred in 50% of patients (≥grade 3 in 13.3%) [51].

CD123 was also targeted by a BiTE, Vibecotamab; the phase I dose-escalation study, enrolling 104 AML, 1 B-cell ALL, and 1 chronic myeloid leukemia (CML), showed, at the higher dose tested (0.75 µg/kg), a cCR rate of 13.7%; CRS occurred in 58.5% of patients (8% grade ≥ 3) [52]. A phase II study was designed in order to investigate the potential role of Vibecotamab in AML patients in first or second morphological remission with detectable MRD and a cut-off expression of CD123 of ≥20% with the primary objective of obtaining MRD negativity [53].

A combination with conventional therapy was tested with a second CD123 × CD3 BiTE: APVO436. This drug was at first evaluated as monotherapy in relapsed/refractory (R/R) AML or MDS. In the AML cohort, 2 of the 34 evaluable patients (5.8%) obtained CR. Overall, IRR/CRS were observed in 50% of patients (≥grade 3 in 13%) [54]. An update of the study evaluating APVO436 as an adjunct to the standard of care was presented at the 2022 ASH meeting. The cCR rate was 20% as monotherapy, 40% in combination with Azacitidine (AZA) and VEN, and 33% in combination with Mitoxantrone, Etoposide, and Cytarabine (MEC) chemotherapy. IRR/CRS were observed in 30% of patients (≥grade 3 in 3.3%) [55].

Recently, new antigen targets are being explored in ongoing clinical trials: Tepoditamab (MCLA-117), a BiTE targeting CD3, and CLEC12A, an antigen present on leukemic stem cells and leukemic blasts whose expression seems stable throughout diagnosis, treatment, and relapse [56], showed, as early clinical results, an ORR of 10.3% with a MLFS rate of 1.7% and a CRS rate of 36.2% (≥grade 3 in 8.6%) [57]; and CLN-049, a FLT3 × CD3 BiTE, which showed high efficacy against human leukemic cell lines and patient-derived AML in mouse xenograft models [58]. No clinical results are currently available (NCT05143996).

### 2.2. NK-Cell-Binding Antibodies

Since the early 2010s, efforts have been made to redirect NK-cell activity against AML blasts. In 2013, Wiernick et al. observed that CD16 × 33 BiKE and ADAM17 inhibitors can overcome inhibitory signals mediated by class I major histocompatibility complex (MHC) recognition and CD16 clipping, resulting in potent NK cell killing of CD33+ AML targets [59]. In the same fashion, Gleason et al. documented a significantly enhanced activation and degranulation of CD16 × CD33 BiKE-induced NK in myelodysplastic syndrome (MDS) cells, irrespective of disease stage and karyotype features [60]. 

More recently, the role of CD16 × CD33 BiKE was investigated in pediatric CD33+ AML and ALL patients. NK cells from the patients profited from restoration of effector functions by BiKE treatment, which, thus, could constitute a promising new option for supporting maintenance therapy or “bridging” consolidation chemotherapy before HSCT [61]. The only clinical results from this class of drugs come from the dose-finding study of GTB-3550, a CD16/IL-15/CD33 TriKE tested in R/R AML and R/R high-risk MDS, presented at the 2020 ASH meeting. These early data in 4 patients did not show an objective response, but NK proliferation and activation were observed [62]. A second report, in 2021, described a 25% ORR without any CR [63] (Table 1). bioengineering-10-01228-t001_Table 1Table 1Results from clinical trials in relapsed/refractory AML-antibody-based therapies.Drug (Phase of the Study)StructureOutcomeYearReferenceFlotetuzumab (Phase I)AntiCD3 × CD123 DARTORR 22%, cCR 19%2018[47]GTB-3550 (Phase I, dose escalation)CD16/IL-15/CD33 TriKENo Response2018[62]AMG673 (Phase I, dose escalation)AntiCD3 × CD33 BiTEORR 22.2%, cCR 3.7%2019[51]AMG330 (Phase I, dose escalation)AntiCD3 × CD33 BiTEcCR 19%2020[50]Vibecotamab (Phase I, dose escalation)AntiCD3 × CD123 BiTEcCR 13.7%2020[52]Tepoditamab (Phase I, dose escalation)AntiCD3 × CLEC12A BiTEORR 10.3% cCR 1.7%2020[57]APVO436 (Phase I, dose escalation)AntiCD3 × CD123 BiTECR 5.8%2021[54]Flotetuzumab (Phase I/II)AntiCD3 × CD123 DARTORR 24%, cCR 20%2021[48]GTB-3550 (Phase I)CD16/IL-15/CD33 TriKEORR 25%2021[63]APVO436 (Phase I)AntiCD3 × CD123 BiTEcCR 20% (monotherapy)cCR 33% (+MEC chemotherapy)cCR40% (+AZA/VEN)2022[55]

## 3. Chimeric Antigen Receptor Strategies

CAR-T cells are constructs composed of an antigen recognition extracellular domain, a transmembrane domain, and an intracellular domain consisting of a co-stimulation domain and cluster of differentiation 3 (CD3 ζ signaling tail) [64] (Figure 2). The variable region of monoclonal antibodies able to bind a target antigen is used as the source for the antigen recognition extracellular domain, which consists of a scFv fragment, VL, and VH chains linked together with short linker peptides of serine-glycine or glutamate–lysine [65]. This novel therapy represented a massive breakthrough in lymphoid malignancies, establishing it as a salvage treatment in ALL, lymphomas, and multiple myeloma [66]. As mentioned above, the path to CAR-T in AML has been rough, but the latest results from the clinical trials provided promising results (Table 2).

### 3.1. CAR-T

The first clinical trial published in 2015, testing a CD33-directed CAR-T, enrolled one R/R AML patient. A decrease in blasts was observed without a clinical response; furthermore, IRR occurred [67].

In 2019, a CAR-T directed against NKG2D (Natural Killer Group 2, Member D), a receptor that transmits activation signals in NK-cells and activates the immune system against NKG2D ligands, expressed in AML blasts and poorly expressed in healthy tissues [68], was tested in seven R/R AML patients. In the same fashion, although a transient improvement of hematologic parameters in one patient was documented, no objective response was observed. No CRS occurred [69].

In 2021, a new trial evaluated a CD33-directed CAR-T in 3 patients; again, none of them met criteria for response to treatment. Two patients developed CRS [70].

In the same year, one case of a CD38+ AML refractory to chemotherapy combined with an anti-CD38 monoclonal antibody (Daratumumab) successfully treated with CD38-directed CAR-T was reported [71]. Although the efficacy was reported in one case, subsequent studies on CART-38 in AML patient-derived xenograft models confirmed the anti-leukemic activity [72]. A CD38-directed CAR-T trial is currently ongoing (NCT04351022).

In the 2022 ASH meeting, several encouraging new data were presented.

PRGN-3006, a CD33-directed CAR-T with IL-15 membrane bound, was tested in 20 R/R AML, 1 chronic myelomonocytic leukemia (CMML) with ≥5% blasts, and 3 high-risk MDS without (cohort 1) or with lymphodepletion (fludarabine 30 mg/m^2^ and cyclophosphamide 500 mg/m^2^ days −5 to −3; cohort 2). No response was observed in cohort 1; among the 10 AML patients enrolled in cohort 2, the cCR rate was 20% with an ORR of 30%. Overall, CRS occurred in 70.8% of patients (≥grade 3 in 4.2%) [73].

UCART123v1.2, an universal, anti-CD123 allogeneic CAR-T product obtained by the ex vivo knockout of the genes of the T-cell receptor (TCR) alpha constant and CD52 to minimize the risk of GvHD and allow the use of anti-CD52-directed therapy as part of lymphodepletion, was tested in combination with Alemtuzumab (anti-CD52) in 16 patients: 1 patient (6.2%) achieved MLFS, and 1 patient had MRD-negative CR. CRS occurred in 93.7% of patients (≥grade 3 in 18.7%) [74].

An anti-CD123 CAR-T was also tested in 12 R/R AML pediatric patients in a phase I dose escalation trial; the ORR was 16.7%, with one patient (8.3%) achieving CR. No CRS occurred [75].

In 2020, the first case of successful CAR-T targeting CLL1 (also named CLEC12A), also targeted by bispecific antibody-based strategies (see above), was reported in a patient with secondary AML [76]. In 2022, anti-CLL1 CAR-T was tested in 10 R/R AML patients with a cCR rate of 70%; furthermore, 6 patients obtained MRD-CR (transient in 4 cases), and 6 patients underwent HSCT. All patients developed CRS; only one patient presented grade ≥ 3 neutropenia [77].

CLL1 was also successfully targeted in the pediatric population; in 2021, a phase I interim analysis showed an ORR of 81.8% with a cCR rate of 72.7% (50% achieving MRD negativity) [78]. In 2022, the phase I/II study confirmed the promising results: 4 out of 8 recruited patients achieved MRD-negative MLFS, 1 patient had MLFS and MRD positivity, 1 patient had CRi and MRD positivity, 1 patient had PR, and 1 patient remained at stable disease (SD) status but had CLL1-positive AML blast clearance [79].

Recently, anti-CLL1 CAR-T with stimulatory domains was tested: 4 patients were treated with CD28/CD27-equipped anti-CLL1 CAR-T, while 3 were treated with 4-1-BB-equipped anti-CLL1 CAR-T. In the first group, 3/4 patients achieved MRD-negative CR, while in the second one, 1 pt achieved MRD-positive CR and one pt achieved MRD-negative CR; CRS was experienced by all patients without ≥grade 3 events [80]. 

Other CLL1-directed CAR-T trials are currently ongoing (NCT04219163, NCT04884984, and NCT04923919).

### 3.2. CAR-NK

To date, early results from only one trial testing CAR-NK in R/R AML patients have been presented.

An anti-CD33 CAR-NK was tested in 10 heavily pre-treated patients (median of lines: 5, range: 3 to 8) in a phase I dose escalation trial (6 × 10^8^, 1.2 × 10^9^, or 1.8 × 10^9^ cells per round after the precondition with Fludarabine (30 mg/m^2^) and Cytoxan 300–500 mg/m^2^ for 3 days to 5 days, determined by tumor burden at baseline).

An MRD-negative CR rate of 60% was achieved. CRS was observed in 70% of patients without ≥grade 3 events [81]. Three clinical trials testing NKG2D-directed CAR-NK are currently ongoing (NCT04167696, NCT04658004, and NCT04623944).bioengineering-10-01228-t002_Table 2Table 2Results from clinical trials in relapsed/refractory AML—CAR-T.Drug (Phase of the Study)StructureOutcomeYearReferenceCD33-CAR-T (Phase I)CAR-T × CD33No response *2015[67]NKG2D-CAR-T (Phase I)CAR-T × NKG2DNo response2019[69]CD33-CAR-T (Phase I)CAR-T × CD33No response2021[70]CAR-T-38 (Phase I)CAR-T × CD38ORR 100% *2021[71]CAR-T-CLL1 (Phase I)CAR-T × CLL1ORR 81.8% cCR 72.7% **2021[78]PRGN-3006 (Phase I, dose escalation)CAR-T × CD33, Il-15 boundORR 0% in non-lymphodepleting cohortORR 30% cCR 20% in lymphodepleting cohort2022[73]CAR-T-123 (Phase I, dose escalation)CAR-T × CD123ORR 16.7%CR 8.3% **2022[75]CAR-T-CLL1 (Phase I)CAR-T × CLL1cCR 70%2022[77]CAR-T-CLL1 (Phase I/II)CAR-T × CLL1ORR 87.5% cCR 75% **2022[79]UCART123v1.2 + Alemtuzumab (Phase I, dose escalation)CAR-T × CD123cCR 12.5%2022[74]CD33-CAR-NK + Fludarabine/Cyclophosphamide (Phase I, dose escalation)CAR-NK × CD33MRD-CR 60%2022[81]CAR-T-CLL1 (Phase I)4-1-BB CAR-T × CLL1CD28/CD27 CAR-T × CLL1CR 67% **CR 75% **2023[80]* One patient was enrolled in the trial. ** Trials enrolling pediatric patients.

### 3.3. Future Directions

Ongoing clinical trials are exploring new targets, such as:CD7 (NCT04762485, NCT04033302), whose targeted CAR-T was able to
efficiently kill CD7+ AML cells and CD7+ primary blasts of R/R-AML patients in vitro and significantly inhibit leukemia cell growth in a xenograft mouse model [82];CD19 (NCT04257175), occasionally expressed in AML blasts (especially in CBF-AML) [83];CD44v6 (NCT04097301), overexpressed in *FLT3* and *DNMT3A*-mutated AML and already successfully targeted by CAR-T constructs in pre-clinical studies [84];CD70 (NCT04662294), expressed on most leukemic blasts but with little or no expression in normal bone marrow samples, is already targeted by the human monoclonal antibody Casatuzumab [85];FLT3 (NCT05023707, NCT03904069, NCT05017883), whose targeted CAR-T showed potent inhibition of leukemia proliferation in xenograft models [86];LILRB4 (alias ILT3) (NCT04803929), highly expressed in AML with monocytic differentiation and suspected to be involved in tumor escape, maintains an immunosuppressive milieu for tumor cells [87];CD276 (NCT04692948), expressed in AML with monocytic differentiation as well, whose targeted CAR-T exhibited efficient antigen-dependent cytotoxicity in vitro and in xenograft models of AML [88]. 

## 4. Checkpoint Inhibitors

The comprehensive knowledge of tumor microenvironment and mechanisms of therapy escape led to the development, since the early 2010s, of drugs targeting the broad landscape of cells and molecules around the tumor cells, first in solid tumors than in hematologic malignancies [89]. In a similar fashion as the other immunotherapies, also for checkpoint inhibitors, lymphoproliferative diseases were the first ones to be successfully treated with this class of drugs [90], while in AML, the results from the clinical trial are less convincing (see Table 3), and the optimal therapy combination and clinical setting are still matters of debate. 

### 4.1. SIRP/CD47 Pathway

CD47 is a transmembrane protein broadly expressed in human cells and on many types of tumors, where it plays a pivotal role in immune escape; in fact, the binding to its ligand, SIRPα (thrombospondin-1 and signal regulatory protein α), triggers a “don’t eat me” signal to the macrophages, inhibiting phagocytosis [91]. In the same fashion, CD47 expression promotes survival by preventing dendritic cell-mediated T-cell necroptosis (Figure 3) [92]. 

In AML, CD47 is overexpressed and correlates with a poor prognosis [93,94]. Majeti et al. showed that blocking monoclonal antibodies directed against CD47 enabled phagocytosis of AML stem cells and inhibited their engraftment in vivo [93]. In a similar way, Jaiswal et al. documented minimal engraftment of AML cells expressing low levels of CD47 in immunodeficient mice; the same cells reached engraftment levels equivalent to those of CD47 high-expressing cells when macrophages were targeted with a SIRPα blocking antibody [95].

A second T cell-dependent mechanism in vivo was documented by Tseng et al., who showed that treatment with an anti-CD47 antibody had a greater effect in wild-type mice with established tumors than in T cell-deficient nude mice with equivalent tumor burden [96].

The first drug targeting SIRP/CD47 interaction was CC-90002, an anti-CD47 monoclonal antibody, whose results were presented in 2019: among the 24 R/R AML patients and 4 high-risk (HR) R/R MDS patients treated, no objective response was documented, with an 82% rate of treatment-emergent adverse events [97].

In the same year, the first results of another anti-CD47 antibody, Magrolimab, in combination with AZA were presented. 25 newly diagnosed (ND) AML patients ineligible for induction chemotherapy were enrolled; among the 16 patients evaluable for response, 11 obtained an objective response (9 cCR, 2 PR), and 5 presented stable disease. Interestingly, 5/6 *TP53*-mutated AML patients achieved cCR. The combination therapy was well tolerated, with a safety profile similar to AZA monotherapy [98].

The 2021 update of the study confirmed the promising results, with a 65% ORR (44% CR, 12% CRi, and 6% MLFS); in the subset of *TP53*-mutated AML, the ORR was 71% (48% CR, 19% CRi, and 5% MLFS) [99]. A phase III trial evaluating Magrolimab plus AZA vs. the physician’s choice of AZA/VEN or intensive chemotherapy in TP53-mutated ND AML (ENHANCE-2 study, NCT04778397) is ongoing.

Surprisingly, the ENHANCE study, testing Magrolimab plus AZA in HR-MDS, was recently discontinued due to futility based on a planned analysis [100]; these results are not currently available.

A trial testing Magrolimab plus Atezolizumab (anti-programmed cell death 1 ligand 1 [PDL1]; see the following chapter) combination therapy (NCT03922477) is ongoing.

Lemzoparlimab is an anti-CD47 antibody endowed with a red blood cell (RBC) sparing property and unique binding epitope, potentially differentiating itself from other CD47 axis-targeting therapies. Preliminary results from the phase I study, presented in 2020, showed a 20% cCR rate (MLFS) with an acceptable safety profile (no patient developed a treatment-related serious adverse event) [101]. The phase II study, along with a phase III study testing Lemzoparlimab in combination with AZA for HR MDS (NCT05709093), are currently ongoing.

In 2022, Evorpacept, a fusion protein containing two engineered high-affinity CD47 binding domains of SIRPα linked to an inactive Fc region of human immunoglobulin was designed to promote the phagocytosis of tumor cells and was tested in combination with VEN and AZA in R/R AML or ND HR AML. A blast decrease was observed in all patients, with an objective response in all ND AML patients and 44.4% of R/R AML patients. Among the 14 patients enrolled, 1 grade 3 CRS occurred [102]. 

### 4.2. PD1/PDL1 Pathway

Programmed cell death protein 1 (PD1) is a cell surface receptor expressed on several types of T cells (Tregs, T-helper, and T-effectors) and other immune cells (B cells, NK cells) and has a fundamental role in countering positive signals through the TCR and CD28 by engaging its ligands PDL1 and/or PDL2 [103]. Thus, signals through the PD1 pathway contribute to regulation of T cell activation and functions, T cell tolerance, and return to immune homeostasis [104,105]. High and sustained expression of PD1 and its ligands is common in various types of cancer, representing a mechanism of immune escape [106]. Multiple evidence has highlighted an upregulation of the PD1/PDL1 pathway in AML and a correlation with poor overall survival, adverse-risk mutations and karyotype anomalies, and a higher relapse rate [107,108,109,110]. 

Several drugs targeting this pathway, blocking PD1 (Nivolumab, Pembrolizumab, and Cemiplimab) or PDL1 (Atezolizumab, Avelumab, and Durvalumab), have been successfully tested and approved by the FDA in solid cancers and hematologic malignancies [89,111], although none of these have been approved for AML.

The first data on Nivolumab in combination with AZA dates back to 2017; the early results, on 53 R/R AML patients, showed an ORR of 35% with a cCR rate of 14%. Furthermore, the median overall survival (5.7 months) was superior to historical survival with AZA-based salvage protocols from the same institution. ≥grade 3 immune-related adverse events (IrAE) were observed in 14% of patients and responded rapidly to steroids [112].

In the same year, the results of the therapy combination with Cytarabine and Idarubicin as first-line therapy in a cohort of ND AML and HR MDS (30 patients with AML 2 with HR MDS) were presented: the cCR rate was 72%. More than grade 3 IrAE occurred in 5 patients [113]. The final results were published in 2019: 44 patients were enrolled (42 patients had AML, 2 had HR MDS, and 4 patients did not receive Nivolumab), and the ORR was 77.2%; at a median follow-up of 17.25 months, the median overall survival was 18.5 months, and the median event-free survival was not reached. Response was consistent across second- and therapy-related AMLs, whereas the *TP53* mutation was more frequent in non-responders (*p* = 0.06). Six patients had seven grade 3–4 IrAEs with an overall acceptable safety profile [114].

Nivolumab was also tested as maintenance therapy. In 2018, it was tested in 14 patients with HR AML (adverse-risk cytogenetics, adverse-risk mutations, MRD+, second remission, therapy-related AML) in CR, with a 6 and 12 month CR duration of 79% and 71%, respectively. Five patents experienced ≥grade 3 IrAE [115]. In 2022, the results of a phase II single-arm trial testing Nivolumab as maintenance therapy in HR AML ineligible for HSCT were published: 15 patients (14 cCR, 1 PR) were enrolled, showing a 6-month relapse-free survival of 57.1% and ≥grade 3 IrAE rates of 26.7% [116]. The results of the REMAIN trial, presented at the 2022 ASH meeting, confirmed the suboptimal results: 80 patients with AML in cCR who were not candidates for HSCT were randomized between maintenance therapy with Nivolumab and observation; maintenance therapy failed to improve overall survival and progression-free survival [117]. 

In a similar fashion, Pembrolizumab was tested in different clinical settings: the combination with AZA was investigated in R/R AML and older ND AML by Gojo et al., who presented the preliminary results at the 2019 ASH meeting. The R/R AML cohort achieved an ORR of 17.2% (cCR 13.8%), while the ND AML cohort achieved 58.8% (cCR 47%). Grade 3/4 IrAE was observed in 9 (24%) patients in the first cohort and 3 (14%) patients in the second one [118]. 

The combination with Decitabine (DEC) was tested in a cohort of 10 R/R AML patients, of whom 3 obtained cCR. The toxicity profile of the combination therapy was consistent with that expected from DEC alone, with the exception of three patients who suffered IrAE, likely as a consequence of Pembrolizumab [119]. 

The combination with chemotherapy was explored in R/R patients by Zeidner et al.: 37 patients were enrolled to receive Pembrolizumab after high-dose cytarabine in a phase II study. The ORR was 46%, with a cCR rate of 38%; the administration of Pembrolizumab was associated with 5 IrAE, and no severe toxicity or delay in hematologic recovery were observed [120]. In a follow-up study, 9 patients undergoing alloSCT were compared with a historical control group of 18 AML patients who underwent alloSCT without prior Pembrolizumab exposure. One-year survival was not significantly different between the treatment groups, whereas 100-day mortality was 0% in the Pembrolizumab group versus 17% in the control group, and there was no increase in grade III-IV acute graft-versus-host disease (GVHD) in patients treated with pembrolizumab prior to alloSCT. No chronic GVHD was seen in patients treated with pembrolizumab prior to alloSCT [121].

Finally, Pembrolizumab was tested as maintenance therapy after autologous HSCT in non-favorable-risk AML patients in the first CR or beyond who were ineligible for alloSCT. Among the twenty patients enrolled, after a median follow-up of 80 months, 14 patients remain alive, with 10 in continuous remission. Treatment was well tolerated, with only 1 non-relapse death [122].

In 2022 the results of a large randomized phase II trial evaluating older patients with AML receiving AZA with or without Darvalumab were published. Although the therapy combination did not raise safety concerns, it failed to achieve a better response (ORR 31.3% vs. 35.4%, median overall survival 13 months vs. 14.4 months) [123]. 

Trials testing anti-PDL1 are currently ongoing (NCT02953561, NCT03395973, NCT02935361, NCT02892318, and NCT315482); no results are available at the moment.

### 4.3. TIM3/GAL-9 Pathway

T cell immunoglobulin and mucin domain 3 (TIM3) is a transmembrane protein expressed on immune cells (T cells, NK cells, and dendritic cells) that binds multiple ligands. One of the most known is Galectin-9 (β-Galactoside-binding lectin 9), which plays an important role in immune cell regulation. Zhu et al. showed that Gal-9, through the TIM3 link, was able to induce intracellular calcium flux, aggregation, and death of T helper cells, and administration of Gal-9 in vivo resulted in selective loss of interferon-gamma-producing cells [124]. In the same fashion, Ndhlovu et al. documented TIM3-dependent negative regulation in NK cells [125], and a higher suppressive function of TIM3+ Treg vs. TIM3-Tregs was observed in the tumor microenvironment [126].

The detection of TIM3 in myeloid malignancies cells, in particular MDS [127] and AML leukemic stem cells [128], along with the discovery of the prognostic negative significance of its overexpression in T cells [129,130,131,132], led to the development of drugs targeting this pathway. Darwish et al. showed that blocking TIM-3 in AML cell lines inhibited cell proliferation [133], whereas Kikushige et al. observed in mice models that anti-TIM3 antibody did not harm reconstitution of normal human HSCs but blocked engraftment of AML after xenotransplantation [134].

The first anti-TIM3 antibody, Sabatolimab, was tested in combination with HMA in a cohort of ND or R/R AML patients. The combination Sabatolimab plus DEC achieved an ORR of 41% in ND AML patients and 25% in R/R AML patients, whereas Sabatolimab plus AZA, tested only in ND AML patients, obtained a 27% ORR. Three IrAE were observed in the DEC cohort vs. none in the AZA cohort [135]. 

In the study update in 2021, in a cohort of 40 evaluable patients with ND AML, the ORR was 40.0%. The response rate was consistent in AML patients bearing an adverse-risk mutation (TP53/RUNX1/ASXL1; ORR: 53.8%). The combination was safe and well tolerated, with grade ≥ 3 adverse events similar to HMA alone [136]. 

Trials evaluating new clinical settings, such as AML in MRD+ CR after HSCT (NCT04623216), and therapy combinations, such as Sabatolimab plus AZA/VEN (STIMULUS-AML1, NCT04150029), are currently ongoing.

### 4.4. CD27/CD70 Pathway

CD70 is a molecule expressed in activated T cells and dendritic cells that, by binding its ligand, CD27, a costimulatory receptor of the tumor necrosis factor superfamily, constitutively expressed on lymphocytes and hematologic stem cells, promotes lymphocyte expansion and cell survival [137,138]. Furthermore, CD70 is also aberrantly expressed by various malignancies, including AML blasts, where it promotes proliferation via the WTN pathway [138].

Thus, as also briefly discussed earlier, the CD27/70 pathway represents a good target to address AML; in fact, Riether et al., in in vitro studies, documented strong proliferation inhibition blocking CD27 and CD70 [138].

Casatuzumab, an anti-CD70, has been tested in combination with AZA in ND AML patients ineligible for chemotherapy in a phase I/II trial. The ORR in patients treated at the phase II dose was 37.9% (all in cCR). Seven patients (18.4%) reported infusion-related reactions, including two with grade 3 events. There were 10 fatal treatment-emergent adverse events; none were considered drug-related [139].

In 2021, the preliminary results of the therapy combination Cusatuzumab plus AZA/VEN in ND AML patients ineligible for chemotherapy were presented: among the evaluable 42 patients, the ORR was 92.9% (CR 47.6%, CRi 57.1%, MLFS 11.9%). Apart from the IrAE (11.4% of patients with 2.3% at grade ≥ 3), the safety profile was consistent with that previously reported for the AZA/VEN regimen [140].

### 4.5. CTLA4/CD80-86 Pathway

CTLA4 (cytotoxic T-lymphocyte-associated protein 4) is a transmembrane protein member of the immunoglobulin gene superfamily expressed by both CD4+ and CD8+ T cells, which, interacting with its ligands, CD80 and CD86, inhibits immune response [141]. Interestingly, CTLA4 is also expressed in AML blast, and its co-expression with PD1 correlates with a poor prognosis [142]. In fact, preclinical studies showed that antibody-based CTLA4 blockade results in an enhancement of the number and activity of T cells with an anti-leukemic effect [143]. 

Ipilimumab, an anti-CTLA4 antibody, was first tested as post-HSCT salvage therapy in various malignancies, including 2 AML patients; no effect was observed [144]. In 2023, Ipilimumab was tested in combination with DEC in 48 patients, split in two cohorts: 25 post-HSCT patients (23 AML and 2 MDS; all R/R) and 23 transplant-naïve patients (15 AML and 8 MDS; including 20 R/R and 3 previously untreated). The cCR rate in the first cohort was 20%, while in the second it was 56.5%. The overall IrAE rate was 44% (11 of 25) in the post-HSCT and 48% (11 of 23) in the transplant-naïve settings [145].

A phase I trial investigating Ipilimumab in combination with donor lymphocyte infusion (DLI) for myeloid malignancies relapsed after HSCT is ongoing (NCT03912064).
bioengineering-10-01228-t003_Table 3Table 3Results from clinical trials of acute myeloid leukemia checkpoint inhibitors.Drug (Phase of the Study)TargetPopulationOutcomeYearReferenceNivolumab + AZA (Phase Ib/II)Anti-PD153 R/R AMLORR 35% cCR 21%2017[112]CC-90002 (Phase I)Anti-CD4724 R/R AML4 R/R HR MDSNo Response2019[97]Nivolumab + Cytarabine + Idarubicine (Phase II)Anti-PD142 ND AML2 HR MDSORR 77.2%2019[114]Pembrolizumab + AZA (Phase II) Anti-PD137 R/R AML22 ND AMLORR 17.2% cCR 13.8%ORR 58.8% cCR 47%2019[118]Magrolimab + AZA (Phase I)Anti-CD4725 ND AMLORR 69% cCR 56%2019[98]Lemzoparlimab (Phase I)Anti-CD475 R/R AMLMLFS 20%2020[101]Sabatolimab + HMA (Phase I)Anti-TIM350 ND orR/R AMLORR 41% (DEC) ORR 27% (AZA)ORR 25% (DEC)2020[135]Pembrolizumab + Cytarabine (Phase II)Anti-PD137 R/RORR 46% cCR 38%2021[120]Sabatolimab + HMA (Phase I/Ib)Anti-TIM348 ND AMLORR 40%2021[136]Magrolimab + AZA (Phase I)Anti-CD4752 ND AMLORR 65% cCR 62%2021[99]Cusatuzumab + AZA/VEN (Phase I)Anti-CD7044 ND AMLcCR 92.9%2021[140]Evorpacept + AZA/VEN (Phase I)Anti-CD4711 R/R AML3 ND HR AMLORR 44.4%ORR 100%2022[102]Pembrolizumab + DEC (Phase II)Anti-PD110 R/R AMLcCR 30%2022[119]AZA ± Darvalumab (Phase II)Anti-PD1129 ND AML randomized to receive AZA with or without DarvalumabORR 31.3% vs. 35.4%2022[123]Cusatuzumab + AZA (Phase I/II)Anti-CD7038 ND AMLcCR 37.9%2023[139]Ipilimumab + DEC (Phase I)Anti-CTLA425 postHSCT (23 R/R AML, 2 R/R MDS)23 HSCT naive (15 AML, 8 MDS; 20 R/R 3 ND)cCR 20%cCR 56.5%2023[145]

### 4.6. Other Immune Checkpoint Targets

*CD200* is a cell membrane glycoprotein of the immunoglobulin supergene family, present on both cells of myeloid/lymphoid origin. Its receptor, CD200R, is expressed on cells of the monocyte/myeloid lineage and some T-cell subsets [146]. CD200/CD200R is an important pathway in immune regulation, and the aberrant expression of CD200 on AML blasts is an efficient mechanism of immune evasion through Tregs expansion and NK cell function suppression [147,148]. On a clinical basis, this translates to a high risk of relapse and an inferior outcome [149,150,151]. In vitro and in vivo studies proved the efficacy and feasibility of targeting this axis in AML [152,153]. A trial testing multiple drugs, including Samalizumab, an anti-CD200 antibody, is currently ongoing (multi-study “Master Protocol [BAML-16-001-M1]”, NCT 03013998).

*LAG-3* (lymphocyte activation gene-3) is a type I transmembrane protein with four Ig-like domains structurally similar to CD4 expressed on activated T cells. The bind with its ligand, MHC II, leads to the inhibition of T-cell functions; in fact, its blockade has been demonstrated to reinvigorate exhausted T cells and strengthen anti-infection immunity [154]. Multiple evidence has shown an overexpression of LAG-3 in T cells in AML [142,155,156]. To address this potential mechanism of immune escape, an ongoing trial is evaluating the therapy combination of Relatlimab, an anti-LAG-3 antibody, with Nivolumab or AZA in R/R AML patients. No results are currently available.

## 5. Conclusions

Immunotherapy is based on mechanisms of immune escape and responsiveness. Such mechanisms can be different in the pre-leukemic and transforming stages of diverse subsets of leukemia. Failure to receive immunotherapy can be triggered by diverse disease etiologies and by chemotherapy resistance and immune evasion. Furthermore, the complex and disease-specific microenvironment may contribute significantly to the differency in efficacy observed between AML and lymphoid disorders [12]. In this regard, the combination of immunotherapies and compounds targeting the leukemic niche could overcome therapy escape [157]. 

Given the genetic and epigenetic heterogeneity of pediatric and adult AML, immunotherapy has still not provided a paradigm-shifting breakthrough. The accumulation of genetic mutations and alterations of pathways defining anti-tumor activity are also very different across subsets of leukemia, which might explain different levels of response. 

However, the safety profile has been good for most of the new drugs, and the therapy was well tolerated by patients, who were commonly elderly and subjected to a heavy previous burden of therapy. In addition, IrAEs were usually low-grade and easily treated with steroid therapy. 

Concerning the anti-leukemic activity, some compounds showed encouraging results, but, unfortunately, there are still only a few randomized comparative studies, which are warranted to clarify the actual benefits and to frame these strategies in the most appropriate clinical subsets.

The ongoing trials evaluating drugs targeting new markers and addressing different patient subsets allow us to envision the possibility, for immunotherapies, of expanding both safe chemo-free strategies and first-line combination therapies in AML.

## Figures and Tables

**Figure 1 bioengineering-10-01228-f001:**
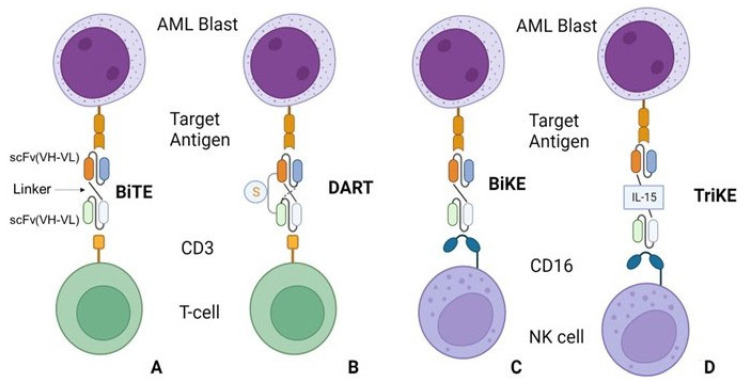
Schematic representation of different bispecific antibody constructs used for cancer immunotherapy. Bispecific antibodies have affinities for both a tumor antigen and an immune cell receptor. (**A**) BiTEs possess two scFv (VH-VL) regions, separated by a flexible linker (arrow), which bind to a tumor antigen and to the CD3 receptor to engage T cells. (**B**) DARTs are BiTEs with an alternative linker and a disulfide bond (denoted by “S”) designed to improve stabilization of the construct. (**C**) BiKEs have the same structure as BiTEs but bind to CD16 instead of CD3 to engage NK cells. (**D**) TriKEs are BiKEs that incorporate IL-15 in the linker domain to intensify the immune response. Abbreviations: AML—acute myeloid leukemia; Bike—bispecific T-cell engager; DARTs—dual affinity retargeting antibodies; BiKe—bispecific killer engager; scFv—single chain variable fragment; TriKE—trispecific killer engager; VH-VL—variable heavy and light chains. The figure was created with Biorender.com.

**Figure 2 bioengineering-10-01228-f002:**
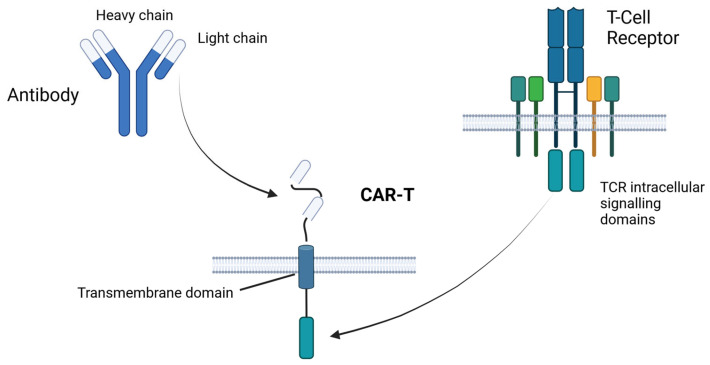
Chimeric antigen receptor T cell (CAR-T) structure. CARs are produced by the fusion of an extracellular antigen-recognition domain, a transmembrane domain, and an intracellular signaling domain. The antigen recognition domain is a scFv (VH-VL) fragment from a monoclonal antibody targeted against a tumor-associated antigen. Abbreviations: scFv—single-chain variable fragment; VH-VL—variable heavy and light chains. The figure was created with Biorender.com.

**Figure 3 bioengineering-10-01228-f003:**
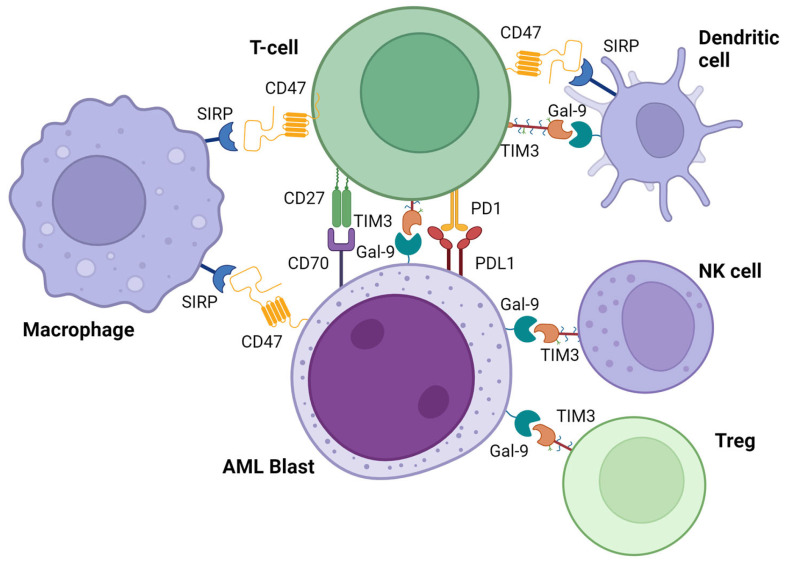
Checkpoint inhibitors in acute myeloid leukemia. Representation of the main immune checkpoint inhibitors. Figure was created with Biorender.com. Abbreviations: AML—acute myeloid leukemia; NK—natural killer; Treg—regulatory T cells.

## Data Availability

Not applicable.

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
