# Peer review of "Immunotherapy in Acute Myeloid Leukemia: A Literature Review of Emerging Strategies"

_bioengineering, 2023, doi:10.3390/bioengineering10101228_

Round 1

Reviewer 1 Report

In this manuscript by Guarnery et al., the authors provide a comprehensive review on immunotherapy for acute myeloid leukemia (AML). The manuscript is well written and organized, and the figures and tables are well prepared. However, there are some comments regarding the manuscript: 1. There is no information about the most popular immunotherapy in AML - allogeneic hematopoietic stem cell transplantation, graft versus leukemia, infusion of donor lymphocytes..... 2. I am missing more detailed information about antibody therapy: - anti-CD33 antibodies, their fractionated dosage regimen and the phase III study ALFA-0701 (its combination with daunorubicin and cytarabine). - anti-CD123 - for the treatment of blast plasmacytoid dendritic cell tumors (BPDCN) and AML. 3. No data regarding vaccine therapy 4. Finally, drugs such as sorafenib, hypomethylating agents, and histone deacetylase inhibitors also have additional immunomodulatory effects.

Author Response

In this manuscript by Guarnery et al., the authors provide a comprehensive review on immunotherapy for acute myeloid leukemia (AML). The manuscript is well written and organized, and the figures and tables are well prepared. However, there are some comments regarding the manuscript: 1. There is no information about the most popular immunotherapy in AML - allogeneic hematopoietic stem cell transplantation, graft versus leukemia, infusion of donor lymphocytes.....

Authors’ response: Thanks for the suggestion. We have now addressed the issue in line 45-55: “Furthermore, other immunologic therapies showed efficacy in AML and some of them are established as gold standard in different settings. The most illustrative example is HSCT: after HSC engraftment, the donor immune system is capable of target the residual leukemic cells, giving life to the so called “Graft-versus-Leukemia” (GvL) effect. Key characters in this setting are lymphocytes, in particular CD4-CD25-FoxP3-positive regulatory T cells (Tregs) which have been proved to avoid alloreactive reactions, prevent reactions of the graft against the host (‘’Graft-versus-host-disease’’ effect, GvHD) and foster engraftment. Other proofs of the immune effects of these cells come from high resolution in-vivo imaging studies, showing co-localization HSC with Tregs, proving thus a role in the creation of a permitting microenvironment, and the efficacy of donor lymphocyte infusion as adoptive immunotherapy for relapse after HSCT.”

  1. I am missing more detailed information about antibody therapy: - anti-CD33 antibodies, their fractionated dosage regimen and the phase III study ALFA-0701 (its combination with daunorubicin and cytarabine). - anti-CD123 - for the treatment of blast plasmacytoid dendritic cell tumors (BPDCN) and AML.

Authors’ response: We added information on current strategies targeting CD33 and CD123 (See line 122-130): “Both Gemtuzumab Ozogamicin and Tagraxofusp represented important breakthroughs in AML treatment: the first one was approved as first-line therapy in combination with daunorubicin and cytarabine in low-intermediate risk de novo AML after phase III ALFA-0701 trial and the metanalysis by Hills et al., which showed a survival advantage in these subsets of patients vs chemotherapy alone. Tagraxofusp was approved for the treatment of blastic plasmacytoid dendritic cell neoplasms (BPDCN) following the trial by Pemmaraju et al., which documented clinical responses in both de-novo and pre-treated patients. Ongoing trials are investigating the efficacy of Tagraxofusp also in other myeloid malignancies, including de novo and secondary AML.”

  1. No data regarding vaccine therapy

Authors’ response: We enriched the introduction with a discussion on vaccine therapy (Line 68-76): “On the other hand, recent immunologic strategies, such as peptide and dendritic cell- based vaccines, didn’t show satisfactory results, especially in active disease settings. Although immunological activity was documented, most of the experienced reported a low rate of clinical responses, with the notable exception of the trial by Dong et al., who showed the efficacy of a combination therapy with low-dose chemotherapy and dendritic cell-based vaccines vs chemotherapy alone in elderly patients. The vaccine efficacy as maintenance therapy in post-chemotherapy and post-HSCT settings provided better results, prolonging survival, and eradicating MRD. Thus, several ongoing phase I and II clinical trials are investigating their potential role in this setting.”

  1. Finally, drugs such as sorafenib, hypomethylating agents, and histone deacetylase inhibitors also have additional immunomodulatory effects.

Authors’ response: We appreciate the suggestion and added additional information about drugs with immunomodulatory effects in line 56-67: “Also other AML treatments present immunomodulatory effects: in preclinical models, Azacytidine (AZA) showed efficacy in preventing GvHD in combination with Tregs, which resulted resistant to the antiproliferative activity of the drug. Furthermore, analysis of bone marrows from patients enrolled in RAS-AZIC trial suggested a AZA-dependent reconstitution of T-cell repertoires. In a similar fashion, histone deacetylase inhibitors (HDACIs) enhance the production and suppressive functions of FOXP3(+) regulatory T cells and stimulate tumor antigenicity and targeted immune system-mediated cytotoxicity CD8+ T cells and NK cells. These results were observed in a suppression of GVHD context while retaining beneficial GVL effect. Sorafenib, a FLT3 inhibitor, through the downregulation of the transcription factor, ATF4, thereby blocking negative regulation of interferon regulatory factor 7, enhances IL-15 transcription in FLT3+ AML cells, which causes an increase in CD8+CD107a+IFN-γ+ T cells, capable of eradicating leukemia cells.”

Reviewer 2 Report

I respect authors' opinion although I do not always agree with their statements. There are a few suggestions for the improvement.

1. Since reviews are generally intended for the broader range of specialists, it is important to explain all abbreviations used in the text. Only a part of those is explained in the text (lines 87-90), but right after these explanations authors use for instance another one IRR/CRS, which is left as is and either explained in other place or not at all. It would be better to do more thorough check and maybe add a paragraph with all abbreviations used. Similarly, abbreviations can be added in the footnotes to the table 1 for better readability.

2. Although the main focus of the review is AML, which has a well documented aging component, authors sporadically refer to ALL and/or pediatric cases, which have very different genetic nature. It would be good to justify more clearly these comparisons and give some comments on a current views why the same treatment works so differently in different cases.

3. Authors mention some treatment cases done with a handful of patients (in one case one patient). It might be a part of the clinical trial, yet it says nothing regarding overall expectation for the broad range clinical use. 

4. I leave it to the authors, but I would avoid emotional expressions and choose more neutral tone.

none

Author Response

I respect authors' opinion although I do not always agree with their statements. There are a few suggestions for the improvement.

Authors’ response: We thank the reviewer for the comment. We highly appreciate the suggestions and we have incorporated them in this revised version of the manuscript.

Since reviews are generally intended for the broader range of specialists, it is important to explain all abbreviations used in the text. Only a part of those is explained in the text (lines 87-90), but right after these explanations authors use for instance another one IRR/CRS, which is left as is and either explained in other place or not at all. It would be better to do more thorough check and maybe add a paragraph with all abbreviations used. Similarly, abbreviations can be added in the footnotes to the table 1 for better readability.

Authors’ response: We made sure that acronyms are spelled the first time and we also added a paragraph with a list of abbreviations used in the manuscript.

  1. Although the main focus of the review is AML, which has a well documented aging component, authors sporadically refer to ALL and/or pediatric cases, which have very different genetic nature. It would be good to justify more clearly these comparisons and give some comments on a current views why the same treatment works so differently in different cases.

Authors’ response: We thank the reviewer. We have added a paragraph in the Conclusions at lines 530-541: “Immunotherapy is based on mechanisms of immune escape and responsiveness. Such mechanisms can be different in pre-leukemic and transforming stages of diverse subsets of leukemia. Failure to immunotherapy can be triggered by the diverse disease etiology and by chemotherapy resistance and immune evasion. Furthermore, the complex and disease-specific microenvironment may contribute significantly in the efficacy differences observed between AML and lymphoid disorders. In this regard, the combination of immunotherapies and compounds targeting the leukemic niche could overcome therapy escape. Given the genetic and epigenetic heterogeneity of pediatric and adult AML, immunotherapy has still not provided  a paradigm-shifting breakthrough. The accumulation of genetic mutations and alterations of pathways defining anti-tumor activity are also very different across subsets of leukemia which might explain different levels of response.

  1. Authors mention some treatment cases done with a handful of patients (in one case one patient). It might be a part of the clinical trial, yet it says nothing regarding overall expectation for the broad range clinical use. 

Authors’ response: We thank the reviewer for the comment. We made sure to mention subsequent studies in the ones reported on a handful of patients. For instance, at lines 243-247.

We agree with the reviewer about the clinical use, but we are also providing a broad overview of all the studies on the topic.

  1. I leave it to the authors, but I would avoid emotional expressions and choose more neutral tone.

Authors’ response: We took in consideration the reviewer’s note. The current version is the result of all reviewers’ comments. We hope that the text reads more neutral now.

Reviewer 3 Report

In the paper “Immunotherapy in Acute Myeloid Leukemia: A literature Review of Emerging Strategies” Guarnera and coworkers describe the development in the field of immunotherapies, such as antibody-based therapy, checkpoint inhibitors and chimeric and antigen receptor strategies in the setting of acute myelogenous leukemia (AML)- They provide a detailed and comprehensive review of potential antigenic targets, available data from pre-clinical, and clinical trials. They utterly discuss the potential future for this treatment approach in AML.

In general, this review is nicely written and discuss these aspects in a clearly perspective. The text is well written and tables are informative. In only have some minor comments.

I think the figures and special the figures legends could be broader in more detail. It is not straight forward for readers not into this specialized field to understand the difference of bite, DART, BiKE and TriKE based on Figure 1 and the figure legend. The same also applied to Figure 2.

In general, I miss a broader discussion about the missing or not fully effective effect of immunotherapy in AML compared, for example, to ALL. What do the authors think may be a future opportunity/challenge when it comes to overcoming these obstacles.

Author Response

In the paper “Immunotherapy in Acute Myeloid Leukemia: A literature Review of Emerging Strategies” Guarnera and coworkers describe the development in the field of immunotherapies, such as antibody-based therapy, checkpoint inhibitors and chimeric and antigen receptor strategies in the setting of acute myelogenous leukemia (AML)- They provide a detailed and comprehensive review of potential antigenic targets, available data from pre-clinical, and clinical trials. They utterly discuss the potential future for this treatment approach in AML.

 In general, this review is nicely written and discuss these aspects in a clearly perspective. The text is well written and tables are informative. In only have some minor comments.

I think the figures and special the figures legends could be broader in more detail. It is not straight forward for readers not into this specialized field to understand the difference of bite, DART, BiKE and TriKE based on Figure 1 and the figure legend. The same also applied to Figure 2.

Authors’ response: We thank the reviewer for the useful and constructive suggestions. We have now included a brief explanation of the content of Figure 1 and Figure 2. We have also modified Figure 1 to facilitate visual interpretation.

We have also modified the manuscript paragraph related to Figure 1 for a better understanding of the bispecific antibody constructs used for cancer immunotherapy:

Line 86-90: “Bispecific antibodies used in cancer immunotherapy have dual affinities targeting both tumor and immune cells, engaging them, and promoting effective antitumoral responses. BiTE antibodies possess two single-chain variable fragment (scFv) regions, each of them with light (VL) and heavy (VH) chains, from two different monoclonal antibodies able to bind a tumor-associated antigen and CD3 receptor in T cells.”

In general, I miss a broader discussion about the missing or not fully effective effect of immunotherapy in AML compared, for example, to ALL. What do the authors think may be a future opportunity/challenge when it comes to overcoming these obstacles.

Authors’ response: We appreciate the reviewers’ suggestion and addressed this point in line 530-537: “Immunotherapy is based on mechanisms of immune escape and responsiveness. Such mechanisms can be different in pre-leukemic and transforming stages of diverse subsets of leukemia. Failure to immunotherapy can be triggered by the diverse disease etiology and by chemotherapy resistance and immune evasion. Furthermore, the complex and disease-specific microenvironment may contribute significantly to the efficacy differences observed between AML and lymphoid disorders. In this regard, the combination of immunotherapies and compounds targeting the leukemic niche could overcome therapy escape.”